# Fluxing of mantle carbon as a physical agent for metallogenic fertilization of the crust

Daryl E. Blanks [1✉], David A. Holwell [1], Marco L. Fiorentini [2], Marilena Moroni [3], Andrea Giuliani[4,5], Santiago Tassara[6,7], José M. González-Jiménez[8], Adrian J. Boyce[9] & Elena Ferrari[3]

Magmatic systems play a crucial role in enriching the crust with volatiles and elements that reside primarily within the Earth's mantle, including economically important metals like nickel, copper and platinum-group elements. However, transport of these metals within silicate magmas primarily occurs within dense sulfide liquids, which tend to coalesce, settle and not be efficiently transported in ascending magmas. Here we show textural observations, backed up with carbon and oxygen isotope data, which indicate an intimate association between mantle-derived carbonates and sulfides in some mafic-ultramafic magmatic systems emplaced at the base of the continental crust. We propose that carbon, as a buoyant supercritical $CO_2$ fluid, might be a covert agent aiding and promoting the physical transport of sulfides across the mantle-crust transition. This may be a common but cryptic mechanism that facilitates cycling of volatiles and metals from the mantle to the lower-to-mid continental crust, which leaves little footprint behind by the time magmas reach the Earth's surface.

[1] Centre for Sustainable Resource Extraction, School of Geography, Geology and the Environment, University of Leicester, University Road, Leicester LE1 7RH, UK. [2] Centre for Exploration Targeting, School of Earth Sciences, ARC Centre of Excellence for Core to Crust Fluid Systems, University of Western Australia, Perth, WA 6009, Australia. [3] Earth Science Department, Milan State University, Milan, Italy. [4] Institute of Geochemistry and Petrology, Department of Earth Sciences, ETH Zurich, Clausiusstrasse 25, Zurich 8092, Switzerland. [5] Kimberlites and Diamonds (KiDs), School of Earth Sciences, University of Melbourne, Parkville, Melbourne, VIC 3010, Australia. [6] Earth and Planetary Sciences, Yale University, PO Box 208109, New Haven, CT 06520-8109, USA. [7] Millennium Nucleus for Metal Tracing Along Subduction, FCFM, Universidad de Chile, Plaza Ercilla 803, Santiago, Chile. [8] Departmento de Mineralogía y Petrología, Universidad de Granada, Facultad de Ciencias, Fuentenueva s/n, 180002 Granada, Spain. [9] Scottish Universities Environmental Research Centre, Rankine Avenue, Scottish Enterprise Technology Park, East Kilbride G75 0QF, UK. ✉email: db404@le.ac.uk

Chalcophile and highly siderophile metals, such as nickel (Ni), copper (Cu), and the platinum-group elements (PGEs), are heavily partitioned together with sulfur (S) into the core and mantle of our planet[1], residing primarily in reservoirs that are inaccessible to direct observation and sampling due to their depth. The metal transfer from the mantle to the crust is facilitated through plumbing systems that transport large volumes of magma up through the lithosphere via interconnected networks of sub-vertical and sub-horizontal conduits, solidified as pipes, dykes, and sills[2]. The subsequent concentration of metals into mineralized bodies requires specific mechanisms, largely dominated by the physical and chemical interaction of sulfide liquids and silicate melts[3].

Distinctive types of mafic and ultramafic magmas derived from the mantle are primed with differing metal budgets according to the composition of the source as well as degree and style of partial melting that generated them. The chalcophile and siderophile metal budget of the mantle itself is largely concentrated in base-metal sulfides. Whereas >20% melting of the mantle leads to their complete exhaustion, yielding komatiitic melts that are generally sulfide undersaturated[4], lower degrees of melting (<15%) generate tholeiitic, mafic alkaline, and subalkaline melts that can be sulfide supersaturated[5,6]. Subsequent transport of metals in ascending melts occurs either as nanoparticles or nanomelts, dissolved ions in the silicate melt, or, primarily, within sulfide liquid droplets[2,3,7]. Metal-rich alloys may be transported alongside sulfide droplets[8] and are entrained within the silicate melt, or collected by the sulfide liquid, as nano- to micrometer-size particles[8–10]. Essentially, this scenario can be viewed as a relatively "dry" magmatic system, with the fluxing of metals across the lithosphere being largely dominated by chemical processes.

However, the mantle rarely attains the degree of partial melting required to completely exhaust the sulfide, platinum-group minerals, and alloy phases that host the metals in the mantle. Therefore, with the notable exception of komatiites[4], the majority of mantle-derived magmas that formed Ni-Cu-PGE sulfide deposits hosted at varying levels in the crust may have been sulfide supersaturated at source. This poses a problem, as dense, metal-rich sulfides are not easily carried upwards in suspension in silicate liquids[11]. Therefore, the question arises as to how these magmas succeeded in transporting their metal cargo upwards from the mantle into the crust.

In the upper crust, hydrous volatile phases have been demonstrated experimentally to provide a viable mechanism of upward physical transport for relatively dense sulfide liquid droplets[12,13] and magnetite crystals[14]. Empirically, this process is reflected in the recent identification of sulfides in magmatic systems associated with coarse-grained hydrous silicate caps[15–17]. In these cases, there is a strong physical attraction of sulfides and oxides to low-density hydrous or saline bubbles or droplets. The low density of the bubble is sufficiently high to overcome the relative density contrast between sulfide/magnetite and the host silicate magma, allowing for effective and potentially rapid upward transport of the metal-bearing phases. This process is analogous to industrial froth floatation, which is used to concentrate dense minerals in ore processing at ambient or relatively low confining pressures. Conversely, such a physical mechanism has not been demonstrated at pressures equivalent to the base of the continental crust, arguably one of the most important parts of the system where sulfide blebs enriched in mantle-derived chalcophile and siderophile metals have to cross a major physical barrier, the Moho discontinuity. However, the process of compound droplet floatation is independent of pressure and thus a plausible, yet thus far unrecognized, mechanism for transporting sulfide in the lower crust should sulfide and volatile super-saturation occur[13].

Here we argue that a process analogous to froth floatation happens in magmatic systems at sub-crustal depths. In this contribution, we explore the possibility that in some mantle-derived, sulfide supersaturated mafic–ultramafic magmas, the physical, and potentially chemical, transport of sulfide liquids may have been facilitated by the presence of immiscible, low-density, $CO_2$-rich volatile phases. This hypothesis is based on the empirical observation that magmatic carbonate has been recognized as a ubiquitous accessory mineral in textural association with sulfide in a number of Ni-Cu-PGE occurrences, particularly those associated with alkaline and/or hydrous magmas[18–21]. These occurrences display very clear and consistent mineralogical characteristics, where magmatic dolomite, calcite, and Fe-Mn-carbonate phases show intimate spatial relationships with Ni-Cu-PGE mineralization. We present a growing body of textural and isotopic evidence that mantle-derived carbonate plays a significant physical role in the mantle to crust fluxing, and upward transport of metals and S in trans-lithospheric magmatic systems.

## Results

**Magmatic sulfide mineralization.** Magmatic sulfide mineralization associated with carbonate as well as accessory apatite and telluride minerals (commonly Ni- and Pt/Pd tellurides) has recently been recognized in several intrusions from the lower and middle crust[19–22]. We document here a series of case studies where a clear textural relationship between carbonate and Ni-Cu-PGE-Te sulfides, plus apatite, can be observed. These natural laboratories present an opportunity to investigate the role of carbon in (C) the transport and accumulation of magmatic sulfides through snapshots in the deep lithosphere.

**Subcontinental lithospheric mantle.** A pristine example of the common association between S and C in the lithospheric mantle is provided by diamonds, wherein sulfides represent the most common inclusion[23]. Sulfides are also widely reported in association with carbonates in metasomatized peridotite xenoliths[24–30]. Here we examine carbonate- and sulfide-bearing mantle xenoliths that sampled the subcontinental lithosherpic mantle (SCLM) beneath: the Deseado Massif auriferous province in Argentinian Patagonia, the Kapvaal Craton in South Africa, and the Apennines in central Italy. In Patagonia, this portion of the SCLM records a protracted history of partial melting and metasomatism produced by multiple episodes of extension and subduction since the Triassic, alongside localized impingement of multiple mantle plumes through time[9,31–33]. We show that the carbonate-sulfide assemblage in these rocks is also commonly associated with accessory Pd-Pt tellurides and F-rich apatite (Fig. 1a–c). Evidence of metasomatism by carbonatite-like melts, which evolved to $CO_2$-rich mafic and later alkaline silicates melts, is preserved in these mantle xenoliths and reflected by the presence of Mg-rich calcite and/or apatite[30]. These phases occur in the form of nodular aggregates associated with Fe-Ni-Cu sulfides, either at the triple junction of primary silicate minerals (Fig. 1a, c) or as inclusions within silicate minerals or glass (Fig. 1b). The Pd-Pt tellurides occur as inclusions in both Fe-Ni- and Cu-rich sulfides (Fig. 1b).

The xenoliths from South Africa are mica-amphibole-rutile-ilmenite-diopside (MARID) rocks that were brought up in kimberlites from the Kimberly area and their petrology has been described elsewhere[34]. They include calcite pools and veins, which contain phlogopite and diopside, plus additional sulfides, apatite, barite, and magnetite (Fig. 1d, e). The calcite pools, which have rounded margins and are present at grain boundary junctions (Fig. 1d), contain a complex assemblage of both Cu-

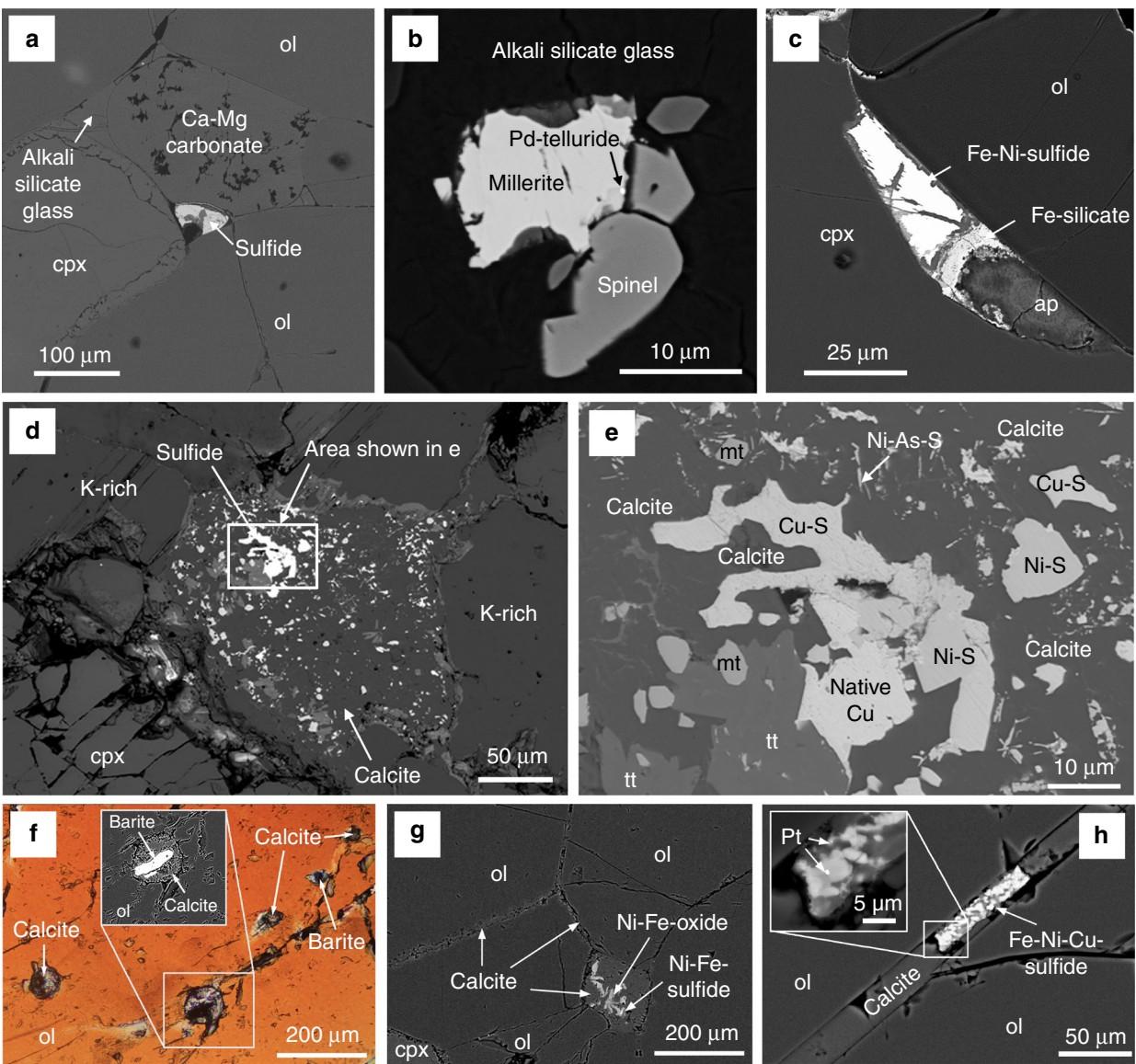

**Fig. 1 Carbonate-sulfide textures in mantle xenoliths from Patagonia, South Africa, and Italy.** All images are backscattered electron images, except **f**, which is a transmitted light image in crossed polars. **a** Sulfide inclusion at the triple junction of Ca-Mg carbonate, olivine (ol), and clinopyroxene (cpx) in a mantle xenolith from Patagonia; **b** glass-hosted sulfide with spinel and Pd-telluride inclusions in millerite in a mantle xenolith from Patagonia; **c** sulfide alongside silicate and apatite (ap) with clinopyroxene and olivine in a mantle xenolith from Patagonia; **d** "pool" of calcite and sulfide within MARID xenolith from Kimberly, South Africa, in contact with K-richterite (K-rich) and diopside (clinopyroxene); **e** enlarged view of sulfide-rich part of the carbonate "pool" shown in **d**, comprising Ni- and Cu sulfides, native Cu, Ni-As-sulfides, magnetite (mt), and titanite (tt); **f** inclusions of calcite, barite, and calcite–barite within an olivine xenocryst from Vulture, Italy; **g** interstitial calcite within mantle peridotite xenolith from Vulture with Ni-Fe-sulfide altered to Ni-Fe-oxides hosted within calcite; **h** Fe-Ni-Cu-sulfide with calcite at the grain boundary between olivine grains in a mantle peridotite from Vulture. A Pt-bearing phase, possibly Pt-S or native Pt, is present within the sulfide.

and Ni-sulfides, titanite, magnetite, and minor native Cu and Ni-As-sulfide (Fig. 1e).

Extrusive carbonatitic lapilli tuffs of the Monticchio Lakes Formation on the Monte Vulture volcano, central Italy, contain mantle xenoliths and xenocrysts (olivine, clinopyroxene, and amphibole) that form the cores of lapilli[35]. The xenoliths have been reported to contain glassy veins containing carbonate, sulfide, and apatite[27], as well as amphibole xenocrysts containing inclusions of carbonate, apatite, and "opaques"[35]. Here we show new data from the olivine xenocrysts that contain rounded globules of calcite with barite (Fig. 1f) and calcite with some associated Ni-Cu-Fe(-Pt) sulfide present as an interstitial phase between olivine grains in wehrlite xenoliths (Fig. 1g, h).

**Lower crust**. Sulfides associated with carbonate as well as accessory apatite and Pt-Pd tellurides have been reported in a number of locations, including the Seiland Igneous Complex, Norway[21,36], and a series of alkaline hydrous and carbonated ultramafic pipes emplaced into the lower continental crust of the Ivrea Zone, Italy. Locmelis et al.[37] and Fiorentini et al.[38] put forward the hypothesis that the Ivrea alkaline pipes and their associated Ni-Cu-PGE sulfide mineralization were emplaced over a protracted time span between ca. 290 and 250 Ma, after the collapse of the Variscan Orogen. The focus here is on the Val-maggia pipe[20,22], where calcite, dolomite, and Fe-Mn-carbonates display different textural and spatial relationships with sulfide mineralization (Fig. 2), which is largely present as nodules,

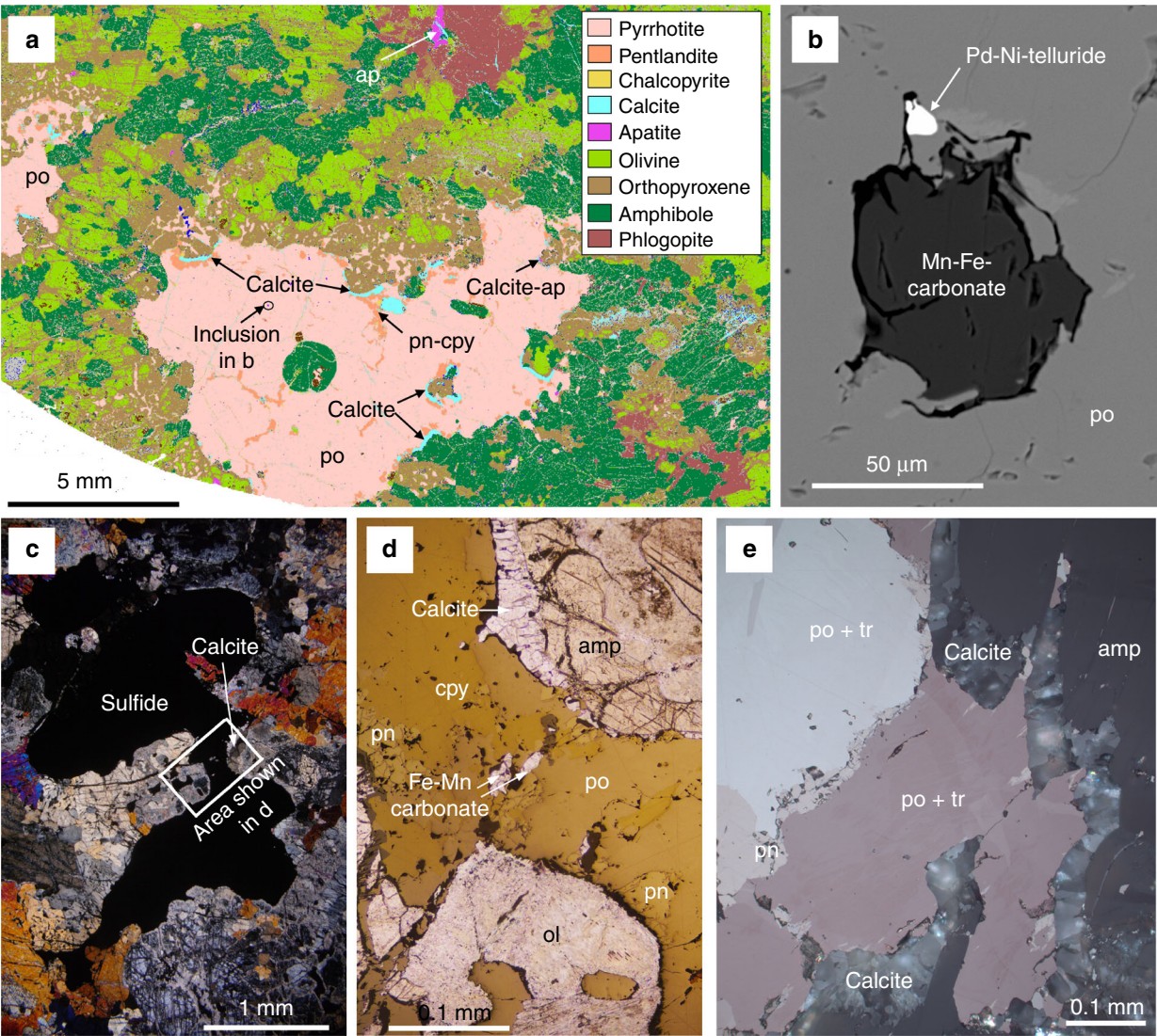

**Fig. 2 Carbonate-sulfide textures in the lower crust at Valmaggia. a** Pyrrhotite (po) with pentlandite (pn) and chalcopyrite (cpy) sulfide blebs with calcite rims hosted within hydrous ultramafic assemblage of phlogopite, orthopyroxene, olivine, and apatite. Mineral map image generated using ZEISS' Mineralogic; **b** backscattered electron image of Mn-Fe carbonate and Pd-telluride inclusion within pyrrhotite shown in **a**; **c** cross-polarized, transmitted light image of sulfide bleb within the Valmaggia alkaline ultramafic rocks with calcite rims around sulfide; **d** enlarged image (reflected and transmitted, plane polarized light) from **c**, showing calcite rims to the sulfide and Fe-Mn-carbonate inclusions; **e** reflected light (cross-polarized) image of crystalline calcite around the margin of a pyrrhotite, troilite (tr), pentlandite sulfide bleb from Valmaggia.

globules, and net-textured assemblages intimately associated with pargasite amphibole and phlogopite[20].

At Valmaggia, Mn-Fe carbonates occur as occasional inclusions (~10 μm) within pyrrhotite blebs (Fig. 2a, b), whereas coarse-grained calcite and Fe-dolomite aggregates show cuspate boundaries along the rims of the sulfide blebs in contact with amphibole, pyroxene, and/or olivine (Fig. 2a, c, d, e). Figure 2e shows the detailed nature of the calcite that forms a convex outer boundary with sulfide. Calcite crystals are aligned perpendicular to the sulfide and silicate surfaces they are in contact with (Fig. 2d). In addition, interstitial dolomite crystals are widely disseminated in the silicate groundmass and Cl-rich apatite is a ubiquitous accessory phase (Fig. 2a). The Ni-Pd-Pt-telluride (melonite) inclusions within Fe-Ni-Cu sulfides characteristically occur towards the margins of the sulfide nodules and generally in close association with calcite/dolomite rims and/or Mn-Fe carbonate inclusions (Fig. 2b).

**Mid crust**. The lamprophyric intrusions of Sron Garbh, Scotland, and the Mordor Alkaline Igneous Complex (MAIC), Northern Territory, Australia, exemplify the association between magmatic Cu-Ni-PGE-Au sulfide mineralization and abundant carbonate in the mid crust[18]. Emplaced within Dalradian metasediments, the Sron Garbh intrusion forms part of the regional magmatic event that post-dated the Caledonian orogeny due to extensional tectonics following slab break-off (ca. 430–408 Ma)[39]. Sulfides at Sron Garbh comprise a Cu-rich, Ni-poor assemblage, which typically occurs as blebby and disseminated chalcopyrite and pyrite with minor millerite and Ni-Co-As sulfides in amphibole cumulates with minor interstitial calcite and apatite[18]. Here we present new textural observations showing that some, although not all, of the calcite occurs as clots that are spatially associated with coarse disseminated chalcopyrite and pyrite (Fig. 3a–c). Palladium-tellurides and bismuthides are also spatially associated with chalcopyrite[18] in the carbonate-sulfide assemblages.

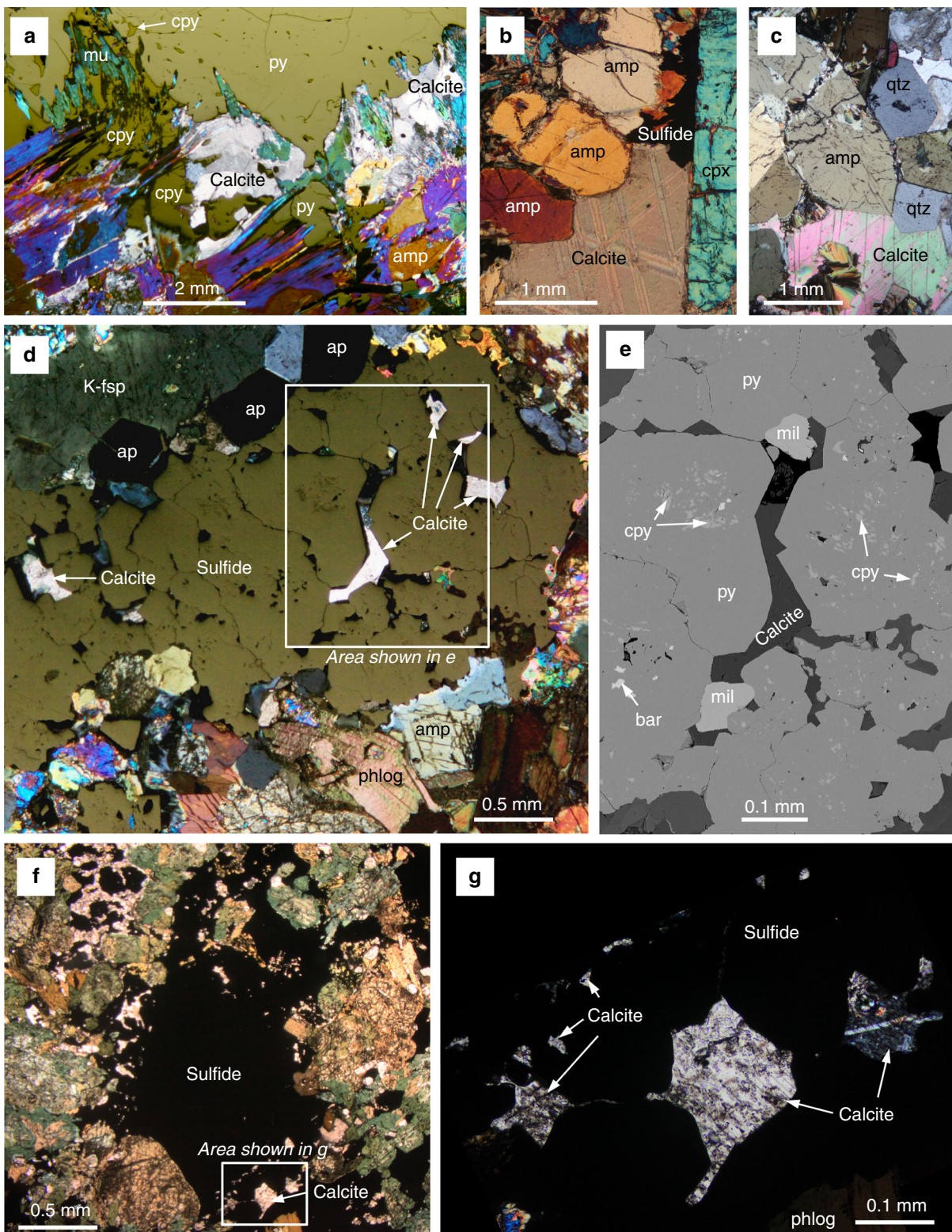

**Fig. 3 Carbonate-sulfide textures in the mid crust. a** Calcite present alongside sulfide from Sron Garbh with amphibole (amp), clinoproxene (cpx), and muscovite (mu) alteration (reflected and transmitted light, crossed polars); **b** calcite alongside sulfide from Sron Garbh (transmitted light, crossed polars); **c** interstitial calcite in lamprophyric cumulate from Sron Garbh with no associated sulfide (transmitted light, crossed polars); **d** sulfide bleb with interstitial calcite in shonkinite from Mordor with phlogopite (phlog), amphibole (amp), K-felsdpar (K-fsp), and apatite (ap) (reflected and transmitted light, crossed polars); **e** enlarged view of area shown in **d** (backscattered SEM image) showing calcite as an interstitial phase to pyrite intergrown with chalcopyrite (cpy), millerite (mil), and minor barite (bar); **f** sulfide bleb with calcite inclusions in shonkinite from Mordor (transmitted light, plane polars); **g** calcite inclusions in sulfide in area shown in **f** (transmitted light, crossed polars).

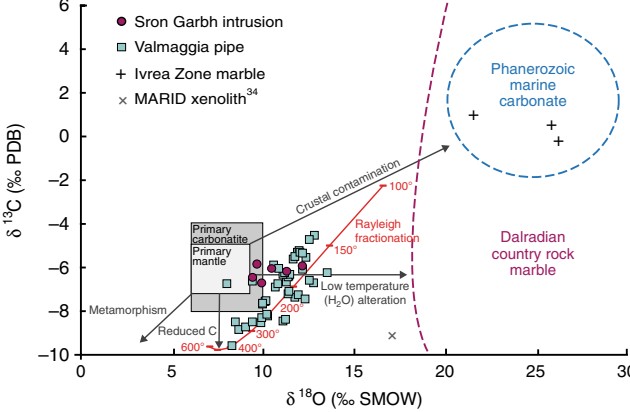

**Fig. 4 Carbon and oxygen isotope compositions of carbonates from mid and lower crustal intrusions (Sron Garbh and Valmaggia), and a mantle xenolith from Kimberly, South Africa[34].** Fields for primary mantle[68] and primary carbonatite[69] are shown. Local country rock marbles are defined by dashed colored lines[70,71]. Isotope compositions of country rock marbles of the Ivrea Zone (this work) fall within the field for Phanerozoic marine carbonates[71]. The red line indicates the composition of calcite crystallized from deuteric fluids at variable temperature (100–600 °C) using the Rayleigh distillation model of Giuliani et al.[43] (see text for details).

The MAIC is a composite intrusion comprising coarse syenite and a mafic–ultramafic body of pyroxenitic cumulates. It is intruded into the high-grade Palaeoproterozoic Arltunga granitic gneisses of the Arunta Orogen, at mid-crustal levels (5–25 km depth)[40]. Magmatic sulfides occur in two associations: blebby sulfides in mafic syenites or "shonkinites"[41], and "reef-like" zones in layered, alkaline ultramafic rocks[42]. Carbonate is present in several forms at Mordor, including veins, dykes, and breccia pipes derived from late-stage $CO_2$-rich fluids and/or residual carbonatitic melts[41]. We present new textural evidence from the mafic syenites that shows an intimate association of calcite as interstitial crystals within blebby pyrite–chalcopyrite (Fig. 3d–g), including accessory Pt-Pd-Ni-tellurides of the moncheite-merenskyite-melonite series.

**Source of carbonates.** The C and O isotopic signatures from carbonate in the magmatic rocks emplaced at various levels in the crust are shown in Fig. 4 (Supplementary Data 1), along with data from selected country rock lithologies and the calcite-bearing MARID xenolith from South Africa[34]. In the MARID sample, sulfides have a mantle-like $\delta^{34}S$ of $-0.7‰$[34], whereas calcite shows a C isotope composition ($\delta^{13}C$ of $-9‰$)[34] that either reflects re-equilibration of metasomatising carbonatitic melts with crustal hydrothermal fluids after kimberlite emplacement[34] or, alternatively, requires involvement of recycled organic C in the source of these melts.

Valmaggia, the lowest crustal occurrence described here, shows little to no evidence for crustal S contamination and sulfides display a magmatic $\delta^{34}S$ signature of $+1.35 \pm 0.25‰$[38]. The associated carbonate phases show variable C-O isotope compositions ($\delta^{13}C = -9.6‰$ to $-4.5‰$; $\delta^{18}O = +8.0‰$ to $+13.5‰$; $n = 45$, this study), with a statistically significant correlation ($R^2 = 0.56$; $n = 44/45$) between $\delta^{13}C$ and $\delta^{18}O$ values (Fig. 4). This correlation could reflect mixing between an end-member with mantle-like $\delta^{18}O$ but isotopically light C isotope composition and a component with isotopically heavy O and C isotopes, such as local Phanerozoic marble country rocks (Fig. 4). However, this interpretation is at odds with the stratigraphic position of marbles in the Valmaggia area, which are located above the intrusion. The most plausible explanation for the isotopic

composition of the Valmaggia carbonates is crystallization from a deuteric (i.e., magmatic) $CO_2$-$H_2O$ fluid at decreasing temperature. Application of the Rayleigh fractionation model of Giuliani et al.[43] shows that the spread in C-O isotope values of the Vamaggia carbonate is reproduced by crystallization from a mantle-derived magmatic fluid with $X_{CO2} = CO_2/(CO_2 + H_2O) = 0.3$, $\delta^{13}C = -6‰$, $\delta^{18}O = +8‰$, at a temperature between 400 and 150 °C (Fig. 4). Carbonate crystallization from a deuteric fluid is consistent with the disequilibrium textures between carbonates, amphibole, and anhydrous silicate phases (i.e., olivine and pyroxene) in the pipe[20].

In the mid-crustal Sron Garbh intrusion, carbonate associated with magmatic sulfide mineralization displays $\delta^{13}C$ of ~6‰ and $\delta^{18}O$ between 9.4‰ and 12.2‰, which are distinct from the local Dalradian country rocks (Fig. 4). Irrespective of lithospheric depth, there is a clear mantle origin of the C in the carbonate crystals that are closely associated with Ni-Cu-PGE sulfide mineralization.

## Discussion

The magmatic carbonate-sulfide occurrences discussed here are consistently hosted by alkaline volatile-rich ultramafic–mafic rocks, which commonly also display accessory P and Te minerals. The intimate association of C- and S-bearing minerals observed in a range of mantle rocks and in mineralized intrusions emplaced within the lower and mid crust may provide insights into the poorly known transport and concentration mechanisms of dense metal-rich sulfides in silicate magmas. We put forward the hypothesis that the observed textural relationships reflect a previously unrecognized process, which may enable the physical fluxing of volatiles and metals across the mantle-crust transition.

In much of the continental and oceanic crust, metals can be transported, (re)cycled, and (re)distributed through typical crustal processes, which almost always include a volatile component (e.g., $H_2O$ and $CO_2$), present as hydrothermal fluids of varying compositions, $fO_2$ and pH. Sulfur and C, largely in the form of carbonate, are common constituents in many hydrothermal ore deposits[44,45] and, although C is clearly present as a volatile in crustal hydrothermal systems[46], it is not generally considered to play a vital role in the mantle-to-crust magmatic transfer of metals, with the exception of rare metals including Nb, Ta, and rare-earth elements in carbonatite magmas.

Nevertheless, the mantle is considered to be an essential reservoir for the global C budget, representing a deep terrestrial reservoir containing $CO_2$, carbonate, diamond and/or Fe-metal carbides[47,48]. In mafic and ultramafic magmas, the major volatile species are $H_2O$, $CO_2$, S, F, and Cl, whereas the metal inventory is dominated by chalcophile and highly siderophile elements (e.g., Ni, Cu, PGE, Au, and Te) in close association with S. In terms of their metal budgets and dominant fluid type(s), the composition of fluids/melts that exsolve from a magma is dependant on the depth (pressure) and the intial composition of the magma, which in turn is a function of the degree of partial melting and the nature of the mantle source[3,49]. The examples of mantle rocks that we show all indicate an intimate association between C- and S-bearing phases (Fig. 1). Isotopic evidence for the association of mantle-derived C with magmatic sulfide occurrences at varying lithospheric levels suggests that the parental melts of the host intrusions were derived from mantle source regions enriched in C (Fig. 4).

In the SCLM, metasomatism by melts and fluids derived from deeper regions of the mantle commonly leads to enrichment of C alongside other incompatible and volatile elements[50]. Carbonate melts exhibit lower viscosities than ambient silicate melts, with high wetting angles resulting in the ability to infiltrate silicate

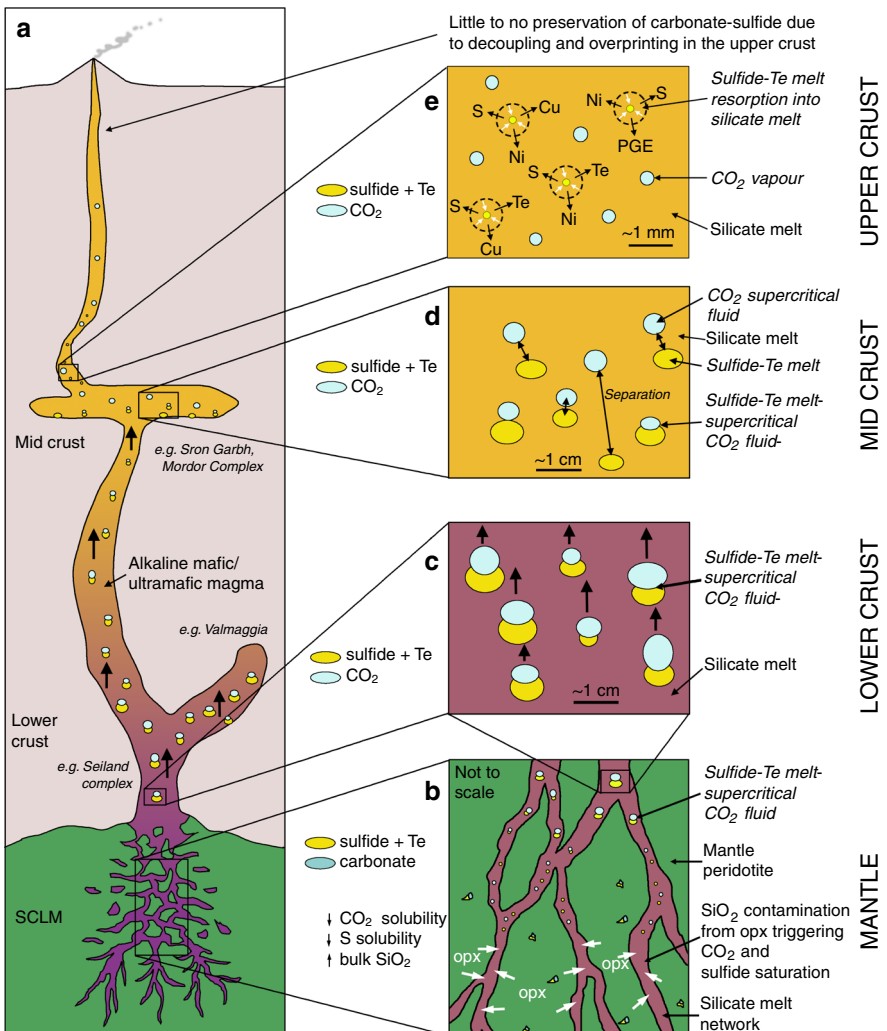

**Fig. 5 Model showing the role of C in transporting metal-rich sulfides from the mantle into the crust. a** Schematic representation of mantle to crust magmatic systems as represented by our case studies. Not to scale; **b** migration of low-degree partial melts reacting with orthopyroxene (opx) causing $SiO_2$ contamination triggering $CO_2$ and sulfide supersaturation; **c** upward migration of metal-rich sulfide melts by physical attraction to droplets of low-density supercritical $CO_2$-rich fluid; **d** mid-crustal separation of supercritical $CO_2$-rich fluid and sulfide melts; **e** mid- to upper crustal redissolution of sulfide into silicate melt and further decoupling from $CO_2$.

minerals and promote widespread lateral metasomatism[51,52]. We therefore concur with previous authors that C and other volatile components (S, Cl, F, and P), along with the Te in the mineralogical assemblages within the host magmas and mantle xenoliths documented here, result from metasomatic enrichment of the SCLM from a range of melts and fluids[30,53,54].

In the deep lithosphere, there appears to be an intimate textural association between Ni-Cu-PGE-Te sulfide mineralization and carbonate. This observation supports the hypothesis that the presence of C may not just be an inherent source characteristic, but that it may also play a critical role in the physical transport and concentration of metal-rich sulfides in ascending magmas. This association appears to be common at varying lithospheric depths ranging from the upper mantle to the mid continental crust, although it is not seen in the upper crust. Some deposits may locally display evidence of carbonate alongside Ni-Cu-PGE sulfide mineralization due to localized assimilation of C-bearing crustal lithologies upon emplacement (e.g., Noril'sk[55]). but this is not the mantle-sourced C that we report from deeper in the lithosphere.

The question is whether the carbonate-sulfide association documented here is simply serendipitous, or if it reflects a C-driven physical mechanism to flux S and metals from the mantle into the lower continental crust, very much like water-dominated processes have been shown to play a fluxing role in the mid to upper crust[12–14]. To address this question, it is necessary to discuss the geochemical behavior of S- and C-bearing fluids in the mantle and in ascending silicate magmas.

The role of S in carrying metals in sulfide liquids is well known[3]. In general, S solubility in silicate melts increases with decreasing pressure[56]. If residual sulfide is present in the mantle during melting, as it would be for melts generated by <10% melting, then the S concentration of the silicate melt should be equal to the S concentration at sulfide saturation[5]. As a result, silicate magmas derived from relatively low degrees of partial melting of the lithospheric mantle may depart their source sulfide supersaturated and, under favorable conditions, remain sulfide supersaturated at the base of the continental crust (see below).

This is evidenced in exposed lower crustal intrusions in the Ivrea Zone, Italy, and the Seiland Complex, Norway[21], which have abundant magmatic sulfides, and also from melt inclusions in Hawaiian basalts that indicate sulfide supersaturation at the mantle-crust boundary[6]. Furthermore, with increased depth sulfide is the dominant S species (over sulfate) at marginally higher

$f\text{O}_2$ conditions[57], such that even though many magmas will be too oxidized to be sulfide supersaturated at upper crustal conditions[58], lower crustal intrusions with similar composition may be supersaturated in sulfide. Regardless, mafic and ultramafic magmas emplaced into the upper crust will most likely be sulfide undersaturated, even if they have previously undergone sulfide saturation. For this reason, externally derived crustal S is considered critical in triggering supersaturation in most upper crustal magmatic systems[59].

Conversely, the role of C in carrying metals is poorly understood. When dissolved in a silicate melt as $CO_2$, C typically becomes less soluble with decreasing pressure[60]. If a silicate melt becomes supersaturated in $CO_2$, any exsolved $CO_2$ should behave as a supercritical fluid rather than a gas phase at temperatures >31 °C and pressures >75 bar (i.e., anything deeper than the uppermost crustal conditions). As a result, the contrasting pressure-dependent solubility of S and $CO_2$ will dictate that if both S and $CO_2$ are present in the same magmatic system, their physical state (e.g., sulfide liquid, S-undersaturated silicate melt, carbonate melt, $CO_2$ supercritical fluid, etc.) will depend on depth and also be a function of the degree of melting and the initial composition of the magmas.

Increasing degrees of mantle melting dilute the concentrations of incompatible elements and volatiles including $CO_2$ and S in melts. As such, melt saturation in $CO_2$ and S, as well as separation of $CO_2$-rich fluids and sulfide liquids in the mantle, are necessarily restricted to low-degree melting regimes (<10%), and therefore alkaline mafic/ultramafic melts. The recent experimental work of Chowdury and Dasgupta[61] on the concentration of S at sulfide saturation in carbonate-rich silicate melts provides a potential theoretical framework in support of our hypothesis. Assimilation of silicate mantle wall rocks ubiquitously affect $CO_2$-rich silicate magmas during their ascent through the lithospheric mantle[62,63]. This process results in an increase of $SiO_2$ contents in alkaline mafic/ultramafic magmas and, therefore, lowers the solubility of $CO_2$, which is inversely related to $SiO_2$ concentrations[60]. At pressure ≥3.5 GPa (~100–110 km of depth), interaction of carbonate-rich melts and peridotite wall rocks (especially orthopyroxene) can drive out large amounts of $CO_2$ from ascending melts and generate $CO_2$-rich supercritical fluids[64]. A large drop in $CO_2$ and related increase in $SiO_2$ contents above 35–40% largely decrease the solubility of reduced S and promotes the formation of immiscible sulfide melts[61] as well. What remains to be addressed is whether or not $CO_2$-rich supercritical fluids and sulfide melts can remain physically connected during ascent once exsolved from their parental silicate magma.

Our proposed model is summarized below and in Fig. 5. In the lithospheric mantle, the carbonate-sulfide(-telluride-apatite) association identified in mantle xenoliths beneath Patagonia, South Africa, and Italy (Fig. 1), alongside examples from the Canary Islands[24], Norway[25], Australia[26], and Scotland[29], highlights the widespread link of carbonate melts and/or $CO_2$-rich fluids associated with sulfide within the metasomatized domains of the lithospheric mantle. The variety of textures and mineralogy shown here (Fig. 1) and in the examples cited above reflects the heterogeneity of the mantle, though in all cases S is intimately associated with C. The common occurrence of sulfides included in diamonds[23] substantiates this widespread link in the mantle.

Experimental studies by Woodland et al.[65] have shown that silicate-carbonatitic melts in the mantle are able to dissolve and transport significant S; however, when mantle-derived magmas are sulfide supersaturated in the deepest portions of the lithosphere, chalcophile metals will be largely transported in sulfide droplets[30,61,66]. Following low-degree partial melting producing carbonate-rich alkaline melts, silica contamination would trigger both $CO_2$ and sulfide supersaturation in the melt (Fig. 5b).

However, supersaturation of sulfide liquids is not conducive to an efficient upward transfer of metals, as sulfides are dense and tend to coalesce into larger blebs, which would settle or break apart[11]. A mechanism is therefore required to overcome this density problem and facilitate the upward physical transport of dense metal-rich sulfide into the crust.

The physical and chemical form of C plays a key role in the efficient transport of sulfide, being most effective when present as a $CO_2$ supercritical fluid phase compared to a carbonate melt, as $CO_2$ can act as a physical buoyancy aid to sulfide droplets. Decarbonation of the $CO_2$-rich silicate melt as a result of interaction with mantle wall rocks in the upper lithospheric mantle (<3.5 GPa) will exsolve $CO_2$[62,64]. At such depths, this will take the form of a $CO_2$-rich supercritical fluid, where the low-density exsolved $CO_2$ fluid phase has a density of ~1.2 g cm$^{-3}$ at pressure of ~2 GPa[62]. The spread in carbonate C-O isotope values observed at Valmaggia supports the involvement of a $CO_2$-rich hydrous fluid (Fig. 4), which could have exsolved from the related melt already at upper mantle depths. Its relatively low density, compared with the silicate magma, will contribute to increasing the inherent buoyancy of the melt, facilitating its rapid ascent and propagation through the Moho discontinuity.

The efficiency of $CO_2$ to transport sulfide liquid will depend on a number of factors, which, by analogue, are all outlined by Yao and Mungall[13] in the context of sulfide transport by water bubbles: the relative volumes and sizes of the volatile and sulfide phases in the compound droplets, and whether they reside in a melt or mush dominated regime. As such, one would expect that the more $CO_2$-rich the melt is (a function of partial melting and source composition), the more efficient its capacity to transport sulfide droplets will be.

The strong wetting behavior of the $CO_2$ fluid phase with the sulfide liquid will significantly increase the buoyancy of metal-rich sulfide liquid droplets. Even if the supercritical $CO_2$ and sulfide melt are immiscible, they nonetheless wet each other (Fig. 5c), as reflected in the textural evidence from Valmaggia (Fig. 2). The relationship of calcite and sulfide shown in Fig. 2e strongly implies that after crystallization of the silicates, supercritical $CO_2$ and sulfide were immiscible liquids wetting each other, with the lower density $CO_2$ forming convex outer boundaries, which were preserved when the sulfide crystallized. The calcite subsequently crystallized from trapped supercritical $CO_2$, as shown by the growth direction of the crystals from both the sulfide and silicate grain boundaries (Fig. 2e). Furthermore, the presence of Mn-Fe carbonate inclusions in the sulfides (Fig. 2e) implies trapping of a C-rich fluid or melt. The calcite and dolomite around the sulfide margins may be the result of reaction of the supercritical $CO_2$ with Ca and Mg from the surrounding silicate melt, whereas the carbonate trapped in the sulfide may have gained Fe and Mn from the surrounding sulfide liquid.

Our proposed "sulfide buoyancy aid" process, operating from the metasomatized lithospheric mantle through to base of the continental crust, is analogous to the established mechanism where aqueous or saline vapor bubbles are suggested to "float" sulfide and/or magnetite at mid-upper crustal depths[12,14,15,17]. However, the critical difference is the deeper lithospheric window where this process operates, which provides a first order mechanism to fertilize the continental crust with mantle-derived chalcophile and siderophile metals. The exsolving $CO_2$ fundamentally changes the physical properties of the ascending magmas, enhancing their bouyancy and catalysing the physical transport of the dense metal-rich sulfide cargoes entrained in mafic–ultramafic melts (Fig. 5a, c). To support this hypothesis, we note that Mungall et al.[12] observed that at the high-pressure end of their experiments (2.5 kbar), the vapor bubbles were

dominated by $CO_2$, which evolve to $H_2O$-dominant at lower pressures. We propose that Valmaggia represents a lower crustal equivalent of the compound model proposed by Mungall et al.[12] for upper crustal systems, with $CO_2$ being the dominant volatile phase as a supercritical fluid.

The very strong spatial relationship between carbonate and sulfide in the lower crustal example at Valmaggia is also present in places in the mid-crustal setting at Mordor and Sron Garbh, where there is also a significant amount of carbonate that is decoupled from the sulfide on a centimeter scale (Figs. 2 and 5d). This change in the C-S association may be due to separation of C and S, which is likely to occur at mid-upper crustal levels (Sron Garbh represents emplacement depths equivalent to at least >1.5 kbar[18]), where the $CO_2$ supercritical fluid that initially fueled the sulfide transport in the silicate melts in the lower crust and mantle may have started to separate, or convert to $CO_2$-$H_2O$ vapor[12] by the time the system reaches the upper crustal levels (Fig. 5d).

Indeed, due to the inverse relationship between pressure and S solubility[56], melts that are supersaturated at depth should start to resorb their sulfide on ascent. The depth at which complete resorption is attained will depend on the initial S concentration of the melt, but is poorly constrained due to a paucity of experimental data between 100 kPa and 1 GPa. However, it is clear that melts become increasingly sulfide undersaturated on ascent, to the point where it is likely that most mid to upper crustal magmas should be sulfide undersaturated. On ascent, sulfide droplets such as those shown in Fig. 5b–d would start to be resorbed (Fig. 5e)[6]. The dissolution of sulfide will return chalcophile metals and S into the silicate melts[54] (Fig. 5e), which would then only be able to form upper crustal magmatic sulfide deposits if sulfide supersaturation is triggered again, e.g., by assimilation of crustal S[59] or prolonged fractional crystallization.

We suggest that the mechanism of $CO_2$-fueled sulfide mobilization not only plays an essential role in the transport of sulfides within the lower-mid crust, but is also critical in the fluxing of metals and sulfide from the mantle itself. We propose that $CO_2$-rich supercritical fluids associated with alkaline mafic–ultramafic magmas enable the initial fluxing of metals and S from the mantle into the crust. Although the C-S association is not uncommon in the mantle and at lower crustal depths, it is rare in most upper crustal settings. The contrasting physio-chemical changes that C and S are subjected to due to decreasing pressure on ascent through the crust (i.e., increasing sulfide solubility vs decreasing $CO_2$ solubility in silicate melts) means that whilst they may be intimately associated as sulfide melt and $CO_2$-rich supercritical fluid (or perhaps sulfide-bearing carbonate melts, e.g., Kogarko et al.[24,28]) in the mantle and lower crust, they are likely to decouple in the upper crust (Fig. 3). This could be the result of a number of pressure-dependent factors, such as redissolution of the sulfide into the silicate melt or degassing of $CO_2$.

The preservation of the intimate spatial C-S association appears to be lost with decreasing crustal depth. Although supercritical $CO_2$ seems to be most critical for the transport of sulfides at mantle and lower crustal conditions, its solubility and low preservation potential increases the likelihood that $H_2O$ or other volatiles may overprint any originally C-driven textural signature and erase any geological record of its former occurrence. In such cases, other phases may appear to be the dominant volatiles preserved within upper crustal sulfide occurrences (e.g., hydrous silicate caps[15]), rather than $CO_2$. The preservation of intimately associated sulfide-carbonate in the upper crust is thus rare and the opportunity for study inherently limited. Exceptions to this may be the upper crustal deposits such as Munali[19], which has been noted to have carbonates associated with sulfides. More generally though, many upper crustal deposits may be the result

of $CO_2$-rich fluids acting as a sulfide buoyancy aid in the lower crust, but the process is untraceable due to either subsequent $CO_2$-sulfide separation, or carbonate overprinting. What our data show are evidence of the fluxing process in action, representing sulfide transport along the lithospheric pathway from source to sink. In summary, we propose that C is a significant agent "in disguise" that may facilitate the transport of sulfides across the mantle-crust transition. We suggest that this may be a common but cryptic mechanism that operates in the deep lithosphere, which leaves very little (if any) footprint behind by the time magmas reach the uppermost crustal levels.

The presence of carbonates alongside sulfides in the SCLM is consistent with a metasomatic origin of these phases and associated metals. The remarkable textural relationships of carbonate as rims and clots alongside sulfide in the lower crust indicate that C, probably as a supercritical $CO_2$-rich fluid, plays a critical role in aiding buoyancy and acting as a driver to propel sulfides up into and through the crust, in a similar way that aqueous and saline vapor bubbles have been proposed to do in the upper lithosphere. The carbonate-sulfide association may decouple at shallower levels, due to the inverse pressure-dependent solubility of S and $CO_2$ in silicate melts, effectively erasing any clue about this important process in the upper crust. The analogy here would be that C acts as the propellant in the first fuel tank that detaches during the launch of a rocket into space. Indeed, it plays a vital role to the success of the departure of the rocket from the Earth's surface (mantle) into the higher levels of the stratosphere (the lower crust). However, evidence of that short but crucially important first step is generally not recorded anywhere by the time the rocket exits the terrestrial atmosphere into space (the upper crust). As such, C acts as the crucial but covert agent in the physical flux of S and metals throughout the lithosphere.

## Methods

**Carbon and oxygen isotopes.** The carbon and oxygen stable-isotope composition of calcite and dolomite from Sron Garbh were analyzed at the Scottish Universities Environmental Research Centre (SUERC) on an Analytical Precision AP2003 mass spectrometer equipped with a separate acid injector system. Measured O isotope ratios are reported as per mil deviations relative to Vienna standard mean ocean water (VSMOW) and C isotopes relative to Vienna PeeDee Belemnite (VPDB) using conventional delta ($\delta$) notation. Mean analytical reproducibility based on replicates of the SUERC laboratory standard NBS-18 (carbonatite) was around ± 0.25% for both carbon and oxygen. The material used for NBS-18 was a carbonatite from Fen, Norway.

Carbon and oxygen isotope analyses of marble, and calcite and dolomite from the Valmaggia ultramafic pipe (Ivrea-Verbano Zone), coupled with carbonate reference materials, were carried out at the Isotope Ratio Mass Spectrometer (IRMS)-stable-isotope laboratory at the Department of Earth Sciences, State University of Milano, Italy. The equipment employed was a ThermoFisher Delta V IRMS coupled with a Finnigan 2 gas bench. Forty carbonate-bearing fragments were obtained from micro-drillcores (Ø < 3 mm) in polished slabs cut from selected highly mineralized samples deriving from the detailed sampling of the Valmaggia mine by Sessa et al.[20]. Materials were purified from sulfides before pulverization in order to avoid erroneous analytical readings due to interferences by $H_2S$. Quantities of material analyzed varied between 0.25 mg (pure carbonates of the international and internal standards) and 0.7 mg (samples with variable fraction of silicates). Carbonate reference materials employed for monitoring the efficiency of the system and the reproducibility of data include calcite standards IAEA-603, IAEA-NBS-18, and IAEA-CO-8, which are of marine (603, Carrara marble) and magmatic (NBS-18 and CO-8, carbonatites from Fen and Kaiserstuhl, respectively) origin. Further control was performed by periodical analysis of additional internal reference materials analyzed in other laboratories and consisting of "refractory" carbonates (dolomite, siderite, and ankerite), thereby covering the spectrum of the carbonate phases occurring in the Valmaggia samples. Powders of samples, international standards, and internal reference samples were placed into borosilicate vials, sealed with butyl rubber septa, and flushed with high-purity helium at 70 °C for 5 min for extracting air. Subsequently, pure anhydrous phosphoric acid was added and acidification of the powder was performed at 80 °C for 12 h before the session of isotopic analysis. The high temperature coupled with the small amount of powder ensures a complete dissolution of refractory carbonates and the absence of fractionation due to incomplete reaction of refractory carbonate species before isotopic analysis. For assuring internal precision and reproducibility, the $\delta^{18}O$ and $\delta^{13}C$ values for each sample were derived from averaging ten individual

measurements, providing an SD < 0.08‰ for standards with pure carbonates and samples particularly carbonate-rich, or higher (up to 0.6‰) for samples with lower carbonate abundance. Data normalization was performed according to the two-point method described in Paul et al.[67] and comparing each "unknown" sample with both international standards and internal reference material affine to the samples. In Supplementary Data 1, $\delta^{13}C$ and $\delta^{18}O$ values are reported using the delta ($\delta$) notation in per mil (‰), relative to VPDB and VSMOW.

## Data availability

Carbon and oxygen isotope data for carbonate minerals from the Valmaggia and Sron Garbh intrusions, and country rocks in the Ivrea Zone, can be found in Supplementary Data 1. The datasets generated and/or analyzed during the current study are available from the corresponding author on reasonable request.

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

## Acknowledgements

This work was financially supported by NERC Minerals Security of Supply (SOS) grant NE/M010848/1 Tellurium and Selenium Cycling and Supply (TeaSe) awarded to the University of Leicester. D.E.B.'s PhD is funded by Consolidated Nickel Mines and the University of Leicester. The study was also funded by the Australian Research Council Centre of Excellence for Core to Crust Fluid Systems (CE11E0070).

## Author contributions

D.E.B., D.A.H., and M.L.F. wrote the manuscript and conceived the idea for a physical role of carbon in the transport of metals across the lithosphere. D.E.B. generated textural and C-O isotopic data from Sron Garbh. M.M. generated C-O isotope data for Valmaggia and contributed to editing the manuscript. A.G. provided images of the mantle xenoliths from South Africa, contributed to writing the revised manuscript and definition of the model, and developed the isotopic model. S.T. provided images from mantle xenoliths from Patagonia and contributed to editing of the manuscript. J.M.G.J. contributed to the writing of the Patagonian xenolith section and to editing of the manuscript. A.J.B. and E.F. contributed to the generation of the C-O isotope analysis at SUERC and Milan, respectively.

## Competing interests

The authors declare no competing interests.
