## [Peer Review File · Nature Communications]

Reviewer #1 (Remarks to the Author):

Dear Editor,

Paper 232486 is discussing the asthenospheric to lithospheric transfer of sulphur and chalcophile/siderophile elements (e.g. Cu, Ni, Au, Pt, Pd) associated with the formation of mafic and ultramafic partial melts.

Essentially, when the asthenospheric mantle experiences partial melting at the order of 20-40 %, the melts will be preferentially enriched (i.e. fertile) in incompatible elements including the PGE's, Cu, Ni, S and C.

During the ascent to upper crustal levels, the fertile melts will experience various events of fractional crystallization, contamination and recharge/mixing with other igneous melts that may facilitate the formation of Cu-Ni-PGE rich sulphide deposits.

According to the author(s), the melts will often be supersaturated with sulphide liquids (i.e. S-Fe melts) in both the melting region in the mantle and at the Moho discontinuity at the mantle-crust transition. Chalcophile elements such as Cu, Ni and the PGE's are strongly partitioning into the sulphide liquids that are considerably more dense than the co-existing silicate liquids. Therefore, they suggest, the sulphide liquids should settle in the deep crust or the Sub Continental Lithospheric Mantle (SCLM) and it would be difficult to explain the genesis of the upper crustal PGE-Cu-Ni sulphide deposits that we de facto observe in nature.

To overcome the transport problem, the author(s) observe that mafic-ultramafic melt products in the lower crust and the SCLM are remarkably enriched in carbonate minerals as well as hydrous minerals when compared to more shallow products of the same melts.

From here they hypothesise that the deep seated mafic ultramafic melts were also supersaturated with supercritical CO₂ fluids. Accordingly, we are both seeing supersaturation with CO₂ and with sulphide liquids co-existing with the silicate melts in the abyssal crust and the SCLM.

Subsequently, they suggest that the CO₂ bubbles, due to their high surface tensions, may actually attach to the drops of sulphide liquids and due to the low density of CO₂, sulphide droplets may be transported across the Moho discontinuity and upwards to higher levels of the crustal column.

The transport of sulphide droplets in mafic magmas is not new, it was first described by Mungall et al. for a shallow volcanic system (c. 200 MPa). What is truly novel in this paper, is that they adapt the principle to the transport of chalcophile elements from the SCLM to the upper crust and suggest that it may be the most important mechanism in asthenospheric to crust transport of Cu-Ni-PGE elements.

Traditionally, it is agreed upon that PGE, Cu Ni and other chalcophile elements are dissolved in the mafic-ultramafic melts (mostly basaltic), transported to the u. crust and, here, magma chamber processes may lead to the formation of sulphide liquids that subsequently scavenge the magma for the the chalcophile elements.

After having read this paper twice I have to admit that it is a very refreshing and thought-provoking contribution to our perception of ore-forming processes in mafic-ultramafic silicate melts. Based on the few studies we have of lower crustal mafic-ultramafic magmatic systems (Ivrea,-Valmaggia Patagonia, Seiland), there is little doubt that we do observe a much higher concentration of carbonates, hydrous minerals and sulphides compared to more shallow system and it is indeed tempting to seek a link between the presence of volatiles and the transport of chalcophile elements. Alone for these reasons it would be interesting to see this paper published – simply to stir some debate and encourage us to rethink what is really happening in the deep seated parts of mafic-ultramafic ore-forming systems.

To facilitate the suggested model we have to have a magmatic conduit system that is supersaturated with sulphide liquids and CO₂ that also transgresses tens of kilometres from the SCLM to higher crustal levels.

This basic requirement is probably also the weakest link in the model for the following reasons:

1. During partial melting in the asthenospheric mantle the mafic ultramafic melts will not be supersaturated with sulphide liquids – they will at best be saturated. Most likely, ultramafic

melts will be undersaturated with sulphide liquid (or simply sulphur) because they form at higher degrees of partial melting of the peridotite assemblage hence, sulphur, partitioning in to the early melt fractions will become diluted at higher degrees of partial melting.

2. During ascent to higher levels throughout the lithosphere, decompression will increase the solubility of sulphur i.e. the melts will be even farther from sulphur supersaturation.
3. However, say that the melts do indeed achieve sulphide supersaturation in the lower crustal magma reservoirs (as clearly seen in Ivrea-Vallmaggia and Seiland) by moderate fractional crystallization and/or other magma-chamber processes. How likely is it that we also have supersaturation with CO₂? After all the solubility of CO₂ at a pressure of 1 GPa may be as much as 1 wt% (less if the concentration of dissolved H₂O is also high).
4. Assuming that CO₂ bubbles do indeed form in the lower crustal reservoirs, the author(s) will of course have their vehicle for sulphide melt transport throughout the upper lithospheric column. However, how far will the sulphide droplets actually ascent? Given that the solubility of sulphide is *increasing* with decreasing pressure, it is reasonable to assume that the sulphide liquids will re-dissolve in the melts; particularly since the melts ought to become more iron-rich higher in the melt column (due to fractional xx in intermittent magma-chambers).
5. The “CO₂ flotation” model is based by the studies of Mungal et al. Mungal is studying a relatively shallow magmatic system where CO₂ was present as a low density super critical gas and low-density CO₂ gasses do indeed have high surface tensions hence can transport heavy phases such as a sulphide liquid. For this reason, in froth flotation they mostly use gas-bubbles. In the lower crust at P around 1 GPa, CO₂, if present, will be a high density super critical liquid with entirely different surface properties. How will this influence the surface tensions and the ability of CO₂ to transport sulphide liquids – I would think it would be greatly reduced?

I hope that the author(s) can address the issues summarised in 1-5, because many of their ideas and the general description of these deep system would be an eye-opener for the geoscience community. However, if they cannot satisfactorily meet these objections, it would be problematic to see it published Nature.

Sincerely yours

Rune B. Larsen

Norwegian University of Science and Technology, Dep. of Geosciences and Petroleum, Trondheim, Norwa

Specific comments to the text (mostly the same as already summarised above but with specific line references):

- 33-34 We may expect that the silicate melts formed by partial melting of the mantle may have high sulphur contents because sulphur is a strongly incompatible element. However, when you are melting >20% of the mantle, the silicate melt will be undersaturated with S, hence you will not form a sulphide liquid at this point in time. As the melts decompress when rising through the mantle the solubility of sulphur will be even higher i.e. we are moving towards more undersaturated conditions. Altogether, both processes prevent the formation of a sulphide liquid.
- 34-35 Are the metal alloys present as immiscible liquids in sulphide liquids or are they actually crystalline phases in the sulphide – silicate melt immiscible system? Either way I find it

very challenging to accept that immiscible sulphide droplets are indeed present at the depth of partial melting in the mantle. After all, at a depth equivalent to e.g. 2 GPa, a picritic melt (for example) may accommodate 1500-3000 ppm of Sulfur. Therefore I doubt that these melts are supersaturated with sulphide unless they have experienced fractional crystallization or have assimilated host-rock sulphide en route to the surface

- 41-44 Generally speaking, the mafic ultra-mafic melts are undersaturated with sulphur throughout the lithospheric column. Saturation occurs when the melts experiences fractional crystallization and the remnant melt becomes over saturated with sulphide or if the melts are contaminated by host-rock sulphate or sulphide.
- 95-96 In this context - what do you mean by metasomatism? Is it infiltration by hydrous-carbonic volatiles or is it infiltration with hydrous-carbonated alkaline melts - for example lamprophyre-carbonatites-kimberlites?
- 196-197 Do you have a reference to support this statement? After all apatite is a very common mineral mafic as well as ultramafic rocks derived from juvenile mantle melts - or perhaps this is what you mean - please clarify.
- 204-205 This is my point regarding apatite - during subduction, you expect to release massive volumes of aqueous fluids - apatite co-existing with these fluids would necessarily become F-rich - i.e. Cl will preferentially partition in to the aqueous fluids (e.g. Candela et al).
- 196-214 196-214 This section is a bit of a convoluted construction and it is quite easy to come up with all sorts of counter arguments to the interpretation of the apatite chemistry. If it is not very important for the aim of this study and it could perhaps be omitted? Altogether - it may be important to know from where Carbon is derived but does it really matter? we know it is down there in the mantle and the SCLM based on multiple studies i.e. you have you vehicle for transporting sulphides don't you?
- 222-224 How can it be justified that the CO₂/S ratio of volcanic gasses mimics the ratios in both the mantle and the parental melts? After all, sulphur partition in to sulphides and carbon partition in to carbonates en route of the melts from the mantle to the surface. I find it very unlikely since it would mean that the magmatic systems, mafi-ultramafic as well as alkaline produce a well balanced mix of sulphides and carbonates independently of the diversity of geological setting experienced in the lithospheric section. How likely is that?
- 230-232 Certainly sulphides forms shortly after emplacement in these systems, but they are not supersaturated - they are simply closer to saturations than systems forming closer to the surface. Essentially, a mantle derived mafic ultramafic melt will reach sulphide supersaturation by perhaps only 20 % fractional crystallization at 1 GPa or so.
- 242 31°C and 75 bar – not 28°C and 100 bar for the critical point of CO₂.

Reviewer #2 (Remarks to the Author):

This work provides a model of metal transportation mechanism from mantle into crust. The flux of mantle carbon is a possible physical agent for this process. The aim of the authors is impressive, but the observation is over-interpreted and the major conclusions are not sufficiently supported by the data.

Lines 43-44: The logic here sounds awkward. It is not necessary that we need an extraordinary set of circumstances to transport metals from mantle into crust. For example, Cu and Au can behave like incompatible elements in an oxidized condition, and were transported through melts into crust. Is this an extraordinary set of circumstances judged by what you say?

The assemblage of carbonate minerals with Fe-Ni-Cu-Pd-Pt alloys or minerals looks interesting in SCLM, lower crust, or mid crust. My question is how you can prove that this is an original assemblage rather than some replacement or overprinting. As you know, sulfides and carbonate minerals are susceptible to later stage melt modification, fluid exsolution, and alteration.

Based on Figure 1, most of your carbonate minerals are quite small (<1 mm). It is not possible to get the pure calcite piece using micro drill for C-O isotopic analysis.

The discussion about Te source is full of speculation. In addition, the presence of F- and/or Cl-rich apatite is not indicative of metasomatised SCLM. The apatite composition is quite susceptible to later-stage magmatic/hydrothermal processes.

Line 235: Actually, most upper crustal intrusions have undergone sulfide saturation process (based on PGE studies from Ian H. Campbell, Jung-Woo Park, and Hongda Hao, GCA, JP, etc.).

Are there any large Cu-Ni-Au deposits related to your study area?

Response to reviews

Fluxing of mantle carbon as a physical agent for metallogenic fertilisation of the crust

Firstly, we thank the editor and reviewers for their constructive comments and time in considering and reviewing our manuscript. We have made some significant revisions to the paper, and here we present our response to the editor and reviewers' comments. The reviewers' comments are pasted here verbatim in black text and our responses are presented in **bold blue** text.

Reviewer #1 Rune Larson

Paper 232486 is discussing the asthenospheric to lithospheric transfer of sulphur and chalcophile/siderophile elements (e.g. Cu, Ni, Au, Pt, Pd) associated with the formation of mafic and ultramafic partial melts.

Essentially, when the asthenospheric mantle experiences partial melting at the order of 20-40 %, the melts will be preferentially enriched (i.e. fertile) in incompatible elements including the PGE's, Cu, Ni, S and C.

During the ascent to upper crustal levels, the fertile melts will experience various events of fractional crystallization, contamination and recharge/mixing with other igneous melts that may facilitate the formation of Cu-Ni-PGE rich sulphide deposits.

According to the author(s), the melts will often be supersaturated with sulphide liquids (i.e. S-Fe melts) in both the melting region in the mantle and at the Moho discontinuity at the mantle-crust transition. Chalcophile elements such as Cu, Ni and the PGE's are strongly partitioning into the sulphide liquids that are considerably more dense than the co-existing silicate liquids. Therefore, they suggest, the sulphide liquids should settle in the deep crust or the Sub Continental Lithospheric Mantle (SCLM) and it would be difficult to explain the genesis of the upper crustal PGE-Cu-Ni sulphide deposits that we de facto observe in nature.

To overcome the transport problem, the author(s) observe that mafic-ultramafic melt products in the lower crust and the SCLM are remarkably enriched in carbonate minerals as well as hydrous minerals when compared to more shallow products of the same melts.

From here they hypothesise that the deep seated mafic ultramafic melts were also supersaturated with supercritical CO₂ fluids. Accordingly, we are both seeing supersaturation with CO₂ and with sulphide liquids co-existing with the silicate melts in the abyssal crust and the SCLM.

Subsequently, they suggest that the CO₂ bubbles, due to their high surface tensions, may actually attach to the drops of sulphide liquids and due to the low density of CO₂, sulphide droplets may be transported across the Moho discontinuity and upwards to higher levels of the crustal column.

The transport of sulphide droplets in mafic magmas is not new, it was first described by Mungal et al. for a shallow volcanic system (c. 200 MPa). **What is truly novel in this paper, is that they adapt the principle to the transport of chalcophile elements from the SCLM to the upper crust and suggest that it may be the most important mechanism in asthenospheric to crust transport of Cu-Ni-PGE elements.**

We would like to highlight this recognition of the novelty of our work. And now, with the revisions, we feel what we present is a much stronger argument, and with more constraints than before.

Traditionally, it is agreed upon that PGE, Cu Ni and other chalcophile elements are dissolved in the mafic-ultramafic melts (mostly basaltic), transported to the upper crust and, here, magma chamber processes may lead to the formation of sulphide liquids that subsequently scavenge the magma for the chalcophile elements.

After having read this paper twice I have to admit that it is a very refreshing and thought-provoking contribution to our perception of ore-forming processes in mafic-ultramafic silicate melts.

We would also like to highlight this positive view on our work that reflects, as we have aimed, the approach of fundamental ore forming processes.

Based on the few studies we have of lower crustal mafic-ultramafic magmatic systems (Ivrea-Valmaggia Patagonia, Seiland), there is little doubt that we do observe a much higher concentration of carbonates, hydrous minerals and sulphides compared to more shallow systems and it is indeed tempting to seek a link between the presence of volatiles and the transport of chalcophile elements. Alone for these reasons it would be interesting to see this paper published – simply to stir some debate and encourage us to rethink what is really happening in the deep seated parts of mafic-ultramafic ore-forming systems.

To facilitate the suggested model we have to have a magmatic conduit system that is supersaturated with sulphide liquids and CO₂ that also transgresses tens of kilometres from the SCLM to higher crustal levels.

We take this point on board and have significantly strengthened the arguments for this scenario by generating a new model that hypothesises supersaturation of sulfide and CO₂ in carbonate-bearing mafic/ultramafic melts in the lithospheric mantle due to the widely reported disequilibrium and related interaction between melt and wall-rock (e.g., Giuliani et al., 2020 Science Advances, and references therein). The following responses to the main five points detail what we have done in response to this.

This basic requirement is probably also the weakest link in the model for the following reasons:

1. During partial melting in the asthenospheric mantle the mafic ultramafic melts will not be supersaturated with sulphide liquids – they will at best be saturated. Most likely, ultramafic melts will be undersaturated with sulphide liquid (or simply sulphur) because they form at higher degrees of partial melting of the peridotite assemblage hence, sulphur, partitioning into the early melt fractions will become diluted at higher degrees of partial melting.

This is true for very high degree mantle melts (e.g., komatiites, tholeiites), but we are generally referring to carbonate-bearing alkaline mafic/ultramafic melts which derive from considerably lower degrees of partial melting (<10%). There is ample evidence from experimental work and melt inclusion work that mantle melts generated under 10% will be supersaturated in sulfide (e.g. Ding and Dasgupta 2018, Weisner et al 2020). We now specifically refer to these melts, and note that higher degree partial melts such as komatiites and picrites may be undersaturated (Keays 1995) as suggested by Dr Larsen. However, several mafic-ultramafic systems that are fertile for

magmatic sulfide deposits are alkaline (as some of our examples are). We have added further references to this effect in both the introductory section (with a rewrite of the second and fourth paragraphs) and the discussion on sulfide supersaturation at depth.

2. During ascent to higher levels throughout the lithosphere, decompression will increase the solubility of sulphur i.e. the melts will be even farther from sulphur supersaturation.

Yes, agreed. However, the recent experimental work of Chowdhury and Dasgupta (2020), which was not published yet at the time of submission, shows that an increase in SiO₂ of CO₂-rich silicate melts beyond 30-40% (i.e. typical SiO₂ contents of primary mantle melts) results in a lower solubility of sulfur. Such an increase in melt SiO₂ is readily achieved via assimilation of mantle wall rock material, a process which appears to be ubiquitous in CO₂-rich mafic/ultramafic melts that ascend through the lithospheric mantle (e.g., Russel et al., 2012 Nature; Giuliani et al., 2020). In other words, in CO₂-bearing melts the maximum solubility of sulfur decreases during ascent as a result of melt evolution which is the opposite of the well-documented increase in S solubility of basaltic melts first documented by Mavrogenes and O'Neill (1999) to which the review here refers. Please note that the experimental work of Chowdhury and Dasgupta (2020) shows no pressure dependence of S solubility in CO₂-rich melts, which is at odds with previous observations for pure silicate melts. This new section has been included in the revised discussion and a revision of panel B on Figure 5.

3. However, say that the melts do indeed achieve sulphide supersaturation in the lower crustal magma reservoirs (as clearly seen in Ivrea-Vallmaggia and Seiland) by moderate fractional crystallization and/or other magma-chamber processes. How likely is it that we also have supersaturation with CO₂? After all the solubility of CO₂ at a pressure of 1 GPa may be as much as 1 wt% (less if the concentration of dissolved H₂O is also high).

We have now added in a section to the discussion that addresses this, with further new references to experimental work. The solubility of CO₂ decreases with pressure and increasing SiO₂ in the melt (Brooker et al., 2011). Therefore, interaction of CO₂-rich mafic/ultramafic melts with mantle wall rocks not only results in lower S solubility but also lower CO₂ solubility in the melt phase. Formation of CO₂-rich fluids during interaction between carbonate melts and peridotites (i.e. orthopyroxene) has been demonstrated experimentally to occur at pressures ≤ 3.5 GPa (Stone and Luth, 2016). We can therefore hypothesise a single process where exsolution of CO₂-rich supercritical fluids in the lithospheric mantle triggers the formation of immiscible sulfide melts due to an increase in SiO₂ contents of the silicate melt phase. In conclusion, our model addresses the linked transport of CO₂ and S from the mantle into the lower crust without calling for speculative magma chamber processes.

4. Assuming that CO₂ bubbles do indeed form in the lower crustal reservoirs, the author(s) will of course have their vehicle for sulphide melt transport throughout the upper lithospheric column. However, how far will the sulphide droplets actually ascent? Given that the solubility of sulphide is increasing with decreasing pressure, it is reasonable to assume that the sulphide liquids will re-dissolve in the melts; particularly since the melts ought to become more iron-rich higher in the melt column (due to fractional xx in intermittent magma-chambers).

Yes this is true, and we thank the reviewer for showing how this part of the story clearly needs to be covered in our discussion. Therefore, we have added a paragraph in the discussion that extends the story into the upper crust with the dissolution of sulfide with decreasing pressure. We have also added a further panel to the final figure to illustrate this. We feel that the story is now more

'complete' although we do emphasise that the role of carbon is most important in the upper mantle to lower crust transition.

5. The "CO₂ flotation" model is based by the studies of Mungai et al. Mungai is studying a relatively shallow magmatic system where CO₂ was present as a low density super critical gas and low-density CO₂ gasses do indeed have high surface tensions hence can transport heavy phases such as a sulphide liquid. For this reason, in froth flotation they mostly use gas bubbles. In the lower crust at P around 1 GPa, CO₂, if present, will be a high density super critical liquid with entirely different surface properties. How will this influence the surface tensions and the ability of CO₂ to transport sulphide liquids – I would think it would be greatly reduced?

Russell et al. (2012) show that exsolved CO₂ at pressures equivalent to the upper mantle has a density of 1.2 g/cm³ – way less than the density of sulfide liquid, or even silicate melt. We use the textural evidence from Valmaggia to show the wetting capability of the CO₂ against the sulfide liquid (Fig. 2). We also add a sentence in this part of the discussion to confirm that the presence of CO₂ will increase the inherent buoyancy of the melt and sulfide droplets.

I hope that the author(s) can address the issues summarised in 1-5, because many of their ideas and the general description of these deep system would be an eye-opener for the geoscience community. However, if they cannot satisfactorily meet these objections, it would be problematic to see it published Nature.

Sincerely yours

Rune B. Larsen

Specific comments to the text (mostly the same as already summarised above but with specific line references):

33-34 We may expect that the silicate melts formed by partial melting of the mantle may have high sulphur contents because sulphur is a strongly incompatible element. However, when you are melting >20% of the mantle, the silicate melt will be undersaturated with S, hence you will not form a sulphide liquid at this point in time. As the melts decompress when rising through the mantle the solubility of sulphur will be even higher i.e. we are moving towards more undersaturated conditions. Altogether, both processes prevent the formation of a sulphide liquid.

This is a good point and we have addressed it. See the points above about us now restricting our model to lower degree partial melts.

34-35 Are the metal alloys present as immiscible liquids in sulphide liquids or are they actually crystalline phases in the sulphide – silicate melt immiscible system? Either way I find it very challenging to accept that immiscible sulphide droplets are indeed present at the depth of partial melting in the mantle. After all, at a depth equivalent to e.g. 2 GPa, a picritic melt (for example) may accommodate 1500-3000 ppm of Sulfur. Therefore I doubt that these melts are supersaturated with sulphide unless they have experienced fractional crystallization or have assimilated host-rock sulphide en route to the surface.

This is the same point as above and our new rewrite addresses it. We also clarify the alloys as 'crystalline phases'.

41-44 Generally speaking, the mafic ultra-mafic melts are undersaturated with sulphur throughout the lithospheric column. Saturation occurs when the melts experiences fractional crystallization and the remnant melt becomes over saturated with sulphide or if the melts are contaminated by host-rock sulphate or sulphide.

This is the same point as above and our new rewrite addresses it. Specially, this work refers to carbonate-rich silicate melts where the pressure dependence of S solubility is low and interaction with mantle rock can increase S solubility due to an increase in SiO₂ content in the melt phase. We have also clarified the statement in the last sentence by explicitly stating 'sulfide supersaturated' melts.

95-96 In this context - what do you mean by metasomatism? Is it infiltration by hydrous carbonic volatiles or is it infiltration with hydrous-carbonated alkaline melts – for example lamprophyre-carbonatites-kimberlites?

We have added citation of the type of metasomatism (from carbonatite-like melts, which evolved to CO₂-rich mafic and later alkaline silicates melts) from the Tassara et al Lithos paper.

196-197 Do you have a reference to support this statement? Afterall apatite is a very common mineral mafic as well as ultramafic rocks derived from juvenile mantle melts - or perhaps this is what you mean - please clarify.

We have now removed this – see response to L196 -214 comment below

204-205 This is my point regarding apatite - during subduction, you expect to release massive volumes of aqueous fluids - apatite co-existing with these fluids would necessarily become F-rich - i.e. Cl will preferentially partition into the aqueous fluids (e.g. Candela et al).

We have now removed this – see response to L196 -214 comment below

196-214 196-214 This section is a bit of a convoluted construction and it is quite easy to come up with all sorts of counter arguments to the interpretation of the apatite chemistry. If it is not very important for the aim of this study and it could perhaps be omitted? Altogether - it may be important to know from where Carbon is derived but does it really matter? We know it is down there in the mantle and the SCLM based on multiple studies i.e. you have your vehicle for transporting sulphides don't you?

This is a very well made point. We have decided to remove most of this section on the origin of the components – as the reviewer says – we know they are there, and in term of C, it is widely available in the mantle and we have our vehicle. Therefore, we now simply have the opening two paragraphs to summarise all this and rebalance our discussion to focus on the physical mechanism of sulfide transport from the mantle.

222-224 How can it be justified that the CO₂/S ratio of volcanic gasses mimics the ratios in both the mantle and the parental melts? Afterall, sulphur partition in to sulphides and carbon partition in to carbonates en route of the melts from the mantle to the surface. I find it very unlikely since it would mean that the magmatic systems, mafi-ultramafic as well as alkaline produce a well balanced mix of sulphides and carbonates independently of the diversity of geological setting experienced in the lithospheric section. How likely is that?

We thank the reviewer's for this point and agree. We have therefore removed all reference to this idea and removed the citation.

230-232 Certainly sulphides forms shortly after emplacement in these systems, but they are not supersaturated - they are simply closer to saturations than systems forming closer to the surface. Essentially, a mantle derived mafic ultramafic melt will reach sulphide supersaturation by perhaps only 20 % fractional crystallization at 1 GPa or so. 242 310C and 75 bar – not 28C and 100 bar for the critical point of CO₂.

Again, this is going back to the question of sulfide supersaturation. We have been more specific here and added in a new statement about the degree of partial melting and referenced Ding and Dasgupta 2018 who show that for melting that leaves residual sulfide, the melts should be saturated. We have added quite a bit of further clarification and discussion to this paragraph.

Reviewer #2 (Remarks to the Author):

This work provides a model of metal transportation mechanism from mantle into crust. The flux of mantle carbon is a possible physical agent for this process. The aim of the authors is impressive, but the observation is over-interpreted and the major conclusions are not sufficiently supported by the data.

We thank the reviewer for his/her comments, and the acknowledgment of our high aims for this paper. We have responded to the individual comments below.

Lines 43-44: The logic here sounds awkward. It is not necessary that we need an extraordinary set of circumstances to transport metals from mantle into crust. For example, Cu and Au can behave like incompatible elements in an oxidized condition, and were transported through melts into crust. Is this an extraordinary set of circumstances judged by what you say?

We have removed this wording and instead posed a question as to how these magmas can transport their metal cargo.

The assemblage of carbonate minerals with Fe-Ni-Cu-Pd-Pt alloys or minerals looks interesting in SCLM, lower crust, or mid crust. My question is how you can prove that this is an original assemblage rather than some replacement or overprinting. As you know, sulfides and carbonate minerals are susceptible to later stage melt modification, fluid exsolution, and alteration.

We argue that these are primary assemblages, based on their textural relationships and also the clearly mantle-like C isotope signatures. In addition, we have increased the images of the textures significantly now by splitting the original figure 1 into three separate figures.

Based on Figure 1, most of your carbonate minerals are quite small (<1 mm). It is not possible to get the pure calcite piece using micro drill for C-O isotopic analysis.

This is incorrect. Silicates contaminating the carbonate will not react with the orthophosphoric acid employed to extract CO₂ from the analysed carbonates (that is, make no isotopic contribution to the evolved CO₂ gas, which is analysed in the mass spectrometer); in any case, a 1-mm grain of carbonate can be readily microdrilled.

The images in Figure 1 are very limited and do not represent the full range of crystal sizes and textures in our samples. We have analysed accurately the C-O isotopes through microdrilling pure carbonate minerals where they have been large enough. The addition of new images to Figure 1 and now the new Figure 2 illustrates the range of grain sizes better.

The discussion about Te source is full of speculation. In addition, the presence of F- and/or Cl-rich apatite is not indicative of metasomatised SCLM. The apatite composition is quite susceptible to later-stage magmatic/hydrothermal processes.

We have now removed this – see response to Reviewer1, L196 comment

Line 235: Actually, most upper crustal intrusions have undergone sulfide saturation process (based on PGE studies from Ian H. Campbell, Jung-Woo Park, and Hongda Hao, GCA, JP, etc.).

Are there any large Cu-Ni-Au deposits related to your study area?

Yes, this is true, but many are no longer saturated. So we have altered our statement here slightly to say “even though they may have previously undergone saturation”. However, the work by Campbell et al is not necessarily applicable to our arguments as they 1) focus on felsic magmas, and 2) consider PGE differentiation into melt in the mantle, rather than saturation on transit.

What we show are not necessarily large deposits, but snapshots of the processes that collude together to form large deposits. It is most likely that in the case of large deposits, the root systems are either not exposed, and thus being able to link the mantle-crust transition directly to an ore deposit in the upper crust is almost impossible.

REVIEWERS' COMMENTS:

Reviewer #1 (Remarks to the Author):

I read the rebuttal to me and the other reviewer's comments by the authors and the revised manuscript. I also read the new paper by Chowdhury and Dasgupta 2020 that they refer to. This is a crucial paper that essentially produces the required "alibi" to imply that indeed ultramafic-mafic mantle-derived melts may reach sulphide-oversaturation at the upper mantle lower crust regime. Their suggestions that the melts upon ascent experience CO₂ oversaturation is based on both common sense and observation of plenty CO₂ inclusions in mantle xenoliths and agree well with our own observations in the deep-seated Seiland Igneous Province.

I notice that the authors have moderated the importance of CO₂ volatile flotation of sulphide droplets to only operate in the upper mantle and the mid-lower crust. That is an essential modification that, given the other improvements, greatly strengthen their theory.

With these essential improvements, I cannot find any significant shortcomings that should stop this important paper from publication. I do no doubt that several geoscientists will be quite excited if this study is published, neither do I doubt that it will meet considerable objections but they logically and persuasively argue their case, and if accepted for publication, I am looking forward to following the debate it may arise.

Only one specific comment: line 270: I see no need to change the phase diagram for CO₂, critical P-T for is 75 bar and 31 oC, not 28oC and 100 bar

Rune B. Larsen

Reviewer #2 (Remarks to the Author):

I have reviewed your revised version. I think most of our comments have been properly addressed.

I still have a few questions.

1. Are there any large Cu-Ni-Au deposits which are likely formed via CO₂ transportation? Geological analogues are needed.
2. Although Chalcophile and highly siderophile metals, such as nickel (Ni), copper (Cu) and the platinum-group elements (PGE), are heavily partitioned into sulfides, we all know the stability of sulfide is closely related to magmatic oxidation state. In most geological settings, such as arcs, the magmas are oxidized with S⁴⁺ and S⁶⁺ as the major phases, and Cu-Au behave like incompatible elements when sulfide is instable. I think CO₂ plays little role in metal transportation in such condition. Therefore, it is quite important to state the optimal conditions for your model.
3. Have you estimated how CO₂ rich in mantle source is efficient for sulfide transportation?

Response to reviews

NCOMMS-19-37996 R1

Blanks et al.

Fluxing of mantle carbon as a physical agent for metallogenic fertilisation of the crust

Firstly, we thank the reviewers for their constructive comments and time in considering and reviewing our revised manuscript and are very pleased to read that our changes have been received extremely positively with only a few outstanding questions to address. We have made the small changes that the reviewers have requested in our second revision (R2). The reviewers' comments to our revised manuscript (R1) are pasted here verbatim in black text and our responses are presented in **bold blue** text.

Reviewer #1 Rune Larson

I read the rebuttal to me and the other reviewer's comments by the authors and the revised manuscript. I also read the new paper by Chowdhury and Dasgupta 2020 that they refer to. This is a crucial paper that essentially produces the required "alibi" to imply that indeed ultramafic-mafic mantle-derived melts may reach sulphide-oversaturation at the upper mantle lower crust regime. **We are delighted to hear this and that the significance of our work is recognised.**

Their suggestions that the melts upon ascent experience CO₂ oversaturation is based on both common sense and observation of plenty CO₂ inclusions in mantle xenoliths and agree well with our own observations in the deep-seated Seiland Igneous Province.

I notice that the authors have moderated the importance of CO₂ volatile flotation of sulphide droplets to only operate in the upper mantle and the mid-lower crust. That is an essential modification that, given the other improvements, greatly strengthen their theory.

We are really pleased to hear this. In the initial submission, Rune set out this as one of five major points we needed to address and so we are happy to see that we have addressed these in full, and that the manuscript is considered to have been strengthened from that.

With these essential improvements, **I cannot find any significant shortcomings that should stop this important paper from publication.** I do no doubt that several geoscientists will be quite excited if this study is published, neither do I doubt that it will meet considerable objections but they logically and persuasively argue their case, and if accepted for publication, I am looking forward to following the debate it may arise.

Again, we are pleased to see that our changes have addressed all the key points raised after the first review, and we agree that the paper is now much stronger following this peer review process. We too look forward to any debate that follows the publication of this!

Only one specific comment: line 270: I see no need to change the phase diagram for CO₂, critical P-T for is 75 bar and 31 oC, not 28oC and 100 bar

We have altered the text accordingly (line 312).

Reviewer #2 (Remarks to the Author):

I have reviewed your revised version. I think most of our comments have been properly addressed.

I still have a few questions.

1. Are there any large Cu-Ni-Au deposits which are likely formed via CO₂ transportation? Geological analogues are needed.

In the final two paragraphs of the discussion, before the concluding paragraph, we make the point that in fact, the C-S association is lost with decreasing crustal depth, and so where we find most deposits in the upper crust, there is no evidence remaining of the role C played. This is why we refer to it as an ‘agent in disguise’ in the final paragraph of the discussion. The point being that we show evidence of a process that shows metal transfer... not necessarily the point of ore deposit formation. So, there may be lots of major deposits that have had their origins in this process, but fingerprinting it in the upper crust is either cryptic or impossible. We put forward one possible example of where this has been preserved – the Munali deposit in Zambia. Furthermore, we show mineralisation (though not economically classed as ‘deposits’) from a number of places in the lower to mid crust.

We have added the following statement in the penultimate paragraph (Around line 439):

“The preservation of intimately associated sulfide-carbonate in the upper crust is thus rare, and the opportunity for study inherently limited. Exceptions to this may be upper crustal deposits such as Munali¹⁶, which has been noted to have carbonates associated with sulfides. More generally though, many upper crustal deposits may be the result of CO₂-rich fluids acting as a sulfide buoyancy aid in the lower crust, but the process is untraceable due to either subsequent CO₂-sulfide separation, or carbonate overprinting. What our data show is evidence of the fluxing process in action, representing sulfide transport along the lithospheric pathway from source to sink.”

2. Although Chalcophile and highly siderophile metals, such as nickel (Ni), copper (Cu) and the platinum-group elements (PGE), are heavily partitioned into sulfides, we all know the stability of sulfide is closely related to magmatic oxidation state. In most geological settings, such as arcs, the magmas are oxidized with S⁴⁺ and S⁶⁺ as the major phases, and Cu-Au behave like incompatible elements when sulfide is instable. I think CO₂ plays little role in metal transportation in such condition. Therefore, it is quite important to state the optimal conditions for your model.

We referred to the work of Matjuschkin et al (2016) which shows that even with high fO₂, at high pressure, sulfide saturation is likely. It is true though, if the conditions are such that the magmas are not sulfide supersaturated, then CO₂ cannot play a role in transporting them. This is actually part of our main model, in that the CO₂-sulfide buoyancy model works in the lower crust, but as we rise up into the upper crust, the sulfides redissolve and the magmas become sulfide undersaturated (this is now in Figure 5E) The reviewer alludes to many upper crustal magmas in arcs (for which our systems is slightly different in any case, being extensional post subduction or intra-cratonic). Nevertheless, we have added a further statement for clarity in the section below around line 300.

“Furthermore, with increased depth sulfide is the dominant sulfur species (over sulfate) at marginally higher fO_2 conditions⁶¹, such that even though many magmas will be too oxidised to be sulfide supersaturated at upper crustal conditions⁶², lower crustal intrusions with similar composition may be supersaturated in sulfide”

In addition, we do state in several places the optimum conditions are <10% mantle melting of a metasomatized mantle source.

3. Have you estimated how CO₂ rich in mantle source is efficient for sulfide transportation?

This is a good question. ‘Efficiency’ is very difficult to quantify and is based on a number of parameters. This can vary a lot. We note how the work of Yao and Mungall shows that at varying ratios of vapour bubble to sulfide droplet, the efficiency of transport is variable. Therefore there are a range of conditions from multiple parameters that can affect ‘efficiency’. We thank the reviewer for making this point and have modified our text to reflect this with this addition of the following short paragraph around line 367:

“The efficiency of CO₂ to transport sulfide liquid will depend on a number of factors, which, by analogue, are all outlined by Yao and Mungall²⁰ in the context of sulfide transport by water bubbles: the relative volumes and sizes of the volatile and sulfide phases in the compound droplets, and whether they reside in a melt or mush dominated regime. As such, one would expect that the more CO₂-rich the melt is (a function of partial melting and source composition), the more efficient its capacity will be to transport sulfide droplets”